Review of the genus Loimia Malmgren, 1866 (Annelida, Terebellidae) from China seas with recognition of two new species based on integrative taxonomy

Wang Weina 1 2 3 4
Sui Jixing 1 2 3 4
Kou Qi 1 2 3 4
Li Xin-Zheng 1 2 3 4 lixzh@qdio.ac.cn
1 Department of Marine Organism Taxonomy and Phylogeny, Institute of Oceanology, Chinese Academy of Sciences , Qingdao , China
2 University of Chinese Academy of Sciences , Beijing , China
3 Center for Ocean Mega-Science, Chinese Academy of Sciences , Qingdao , China
4 Laboratory for Marine Biology and Biotechnology, Qingdao National Laboratory for Marine Science and Technology , Qingdao , China
Reimer James
Electronic publication date: 2020 Jul 16
Publication date: 2020
Volume: 8
Electronic Location ID: e9491
Received 2019 Dec 21; Accepted 2020 Jun 16
Copyright: © 2020 Wang et al.
Copyright year: 2020
Copyright holder: Wang et al.
License: This is an open access article distributed under the terms of the Creative Commons Attribution License, which permits unrestricted use, distribution, reproduction and adaptation in any medium and for any purpose provided that it is properly attributed. For attribution, the original author(s), title, publication source (PeerJ) and either DOI or URL of the article must be cited.
License URL: https://creativecommons.org/licenses/by/4.0/

Keywords: Polychaeta, Loimia, New species, Marine biodiversity, Integrative taxonomy, China Sea

Funding: National Natural Science Foundation of China 31872194 China Ocean Mineral Resources Research and Development Association Program DY135-E2-3-04 Science & Technology Basic Resources Investigation Program 2017FY201404 Chinese Academy of Sciences XDA23050304 Qingdao National Laboratory for Marine Science and Technology 2015ASKJ01 This work was supported by the National Natural Science Foundation of China (No. 31872194) and the China Ocean Mineral Resources Research and Development Association Program (DY135-E2-3-04) the Science & Technology Basic Resources Investigation Program of China (No. 2017FY201404); the Strategic Priority Research Program of the Chinese Academy of Sciences (XDA23050304); the Scientific and Technological Innovation Project Financially Supported by Qingdao National Laboratory for Marine Science and Technology (No. 2015ASKJ01). The funders had no role in study design, data collection and analysis, decision to publish, or preparation of the manuscript.

==============================
Specimens of the genus Loimia (Annelida, Terebellidae) deposited in the Marine Biological Museum of the Chinese Academy of Sciences (MBMCAS) together with materials newly collected from China seas were examined. Based on morphological comparisons and molecular analysis, some specimens collected from the coasts of Shandong province and Guangxi province were confirmed as two new Loimia species respectively (Loimia borealis sp. n. and Loimia macrobranchia sp. n.). Morphologically, L. borealis sp. n. is distinguished from previously known species of this genus by having seven equal sized ventral shields, with length five times the width; this species was retrieved as sister to the clades of Loimia arborea Moore, 1903 and Loimia bandera Hutchings, 1990 in the phylogenetic tree, which was reconstructed based on mitochondrial COI gene. Loimia macrobranchia sp. n. differs from congeners by the large size of its first pair of branchiae with a thick main stem and about 18 dendritic branches arranged in two levels. A key to identifying Loimia species found in Chinese seas is given.

Introduction

Terebellids are common sedentary polychaetes in marine bottoms, found worldwide from shallow water to deep-sea environments (Rouse & Pleijel, 2001). They are selective deposit feeders using long, grooved and ciliated buccal tentacles to collect sediment and organic matters (Fauchald & Jumars, 1979; Jumars, Dorgan & Lindsay, 2015).

Terebellidae Johnston, 1846 is the largest family of Terebelliformia. A recent review of Terebellidae Johnston, 1846 has been undertaken by Hutchings, Nogueira & Carrerette (2017). According to Hutchings, Nogueira & Carrerette (2017), Terebellidae has 44 genera currently, several of which are monotypic. The genus Loimia Malmgren, 1866 comprises 29 valid species occurring worldwide, and most of them are found in tropical areas (Lavesque et al., 2017). Live specimens of Loimia, living in tubes, are usually large and brightly colorful, with long, grooved tentacles and blood-red regions at the termination of mid-ventral shields (Nogueira, Hutchings & Carrerette, 2015).

According to Liu (2008), five species have been documented from Chinese seas: Loimia arborea Moore, 1903; Loimia ingens (Grube, 1878); Loimia medusa (Savigny, 1822); Loimia bandera Hutchings, 1990 and Loimia montagui (Grube, 1878). Loimia arborea was described from Suruga Bay in Japan (Moore, 1903). Loimia bandera was described from Hong Kong (Hutchings, 1990). Loimia montagui and Loimia ingens was described by Grube (1878) from the Philippines. Hutchings & Glasby (1988) examined the materials of L. ingens from Australian waters and suggested that L. ingens should be treated as a species complex. Afterwards, Hutchings (1990) checked materials from Hong Kong and suggested it seemed preferable to refer the Hong Kong material to the Loimia ingens species complex. He also recommended that more material from the Indo-Pacific was needed to be examined before splitting up the complex. Loimia medusa, the type species of the genus Loimia, was described by Savigny (1822) but the original descriptions lacked detailed illustrations of diagnostic characters and type material has never been deposited in any museum (Carrerette & Nogueira, 2015). Hutchings & Glasby (1988) referred specimens previously identified as L. medusa from Australia to a new species, and suggested that L. medusa does not occur in Australian waters. Subsequently they redescribed L. medusa based on materials collected close to the type locality and assigned a neotype. They also suggested that the species was restricted to the Arabian Sea region and in many cases reports from other distant localities might be misidentification (Hutchings & Glasby, 1995).

Recently, some specimens of Terebellidae were collected during two surveys in Shandong and Guangxi provinces in China. After observation, these specimens were confirmed as Loimia species with lobes on segments 1 and 3, three pairs of branchiae on segments 2–4, rectangular mid-ventral shields from segments 2 to the posterior region where notopodia terminate, 17 pairs of notopodia beginning from segment 4, neuropodia beginning from segment 5, bearing pectinate uncini arranged in single rows on segment 5–10 and in double rows on segments 11–20. Since these Loimia specimens were never described before, we herein formally describe them as new species to science. In addition to morphological analyses, we performed a barcoding analysis using sequences of the mitochondrial COI gene to help the identification of specimens.

Materials and Methods

Sample collection and morphological examination

Specimens of Loimia were collected from coastal areas of Chinese seas (Table 1). The collected specimens were anesthetized with 7% magnesium chloride (MgCl2) and photographed using a Canon EOS 600D camera. Thereafter they were preserved in 75% ethanol. All specimens were deposited in the MBMCAS. Specimens were observed, measured, and photographed with a Nikon SMZ25 stereomicroscope. The detailed morphology of the lobes, notochaetae, and uncini was observed and documented with a Nikon Ni-U stereomicroscope and a Hitachi S-3400N scanning electron microscope (SEM).

Table 1 Specimens information.

Species used in the molecular analysis with reference to their voucher ID, specimen location, GenBank accession numbers, GPS coordinates and references.

Species	Location	Vouchers	GenBank accession number	GPS coordinates	References	
Loimia borealis sp. n.	China, Shouguang City	MBM286591	MN133237	37°16’34.00"N	This study	
MBM286593	MN133238	119°02’19.44"E	This study	
MBM286592	MN133239		This study	
MBM286585	MN133240		This study	
MBM286586	MT246207		This study	
MBM286587	MT246208		This study	
Loimia macrobranchia sp. n.	China, Fangchenggang City	MBM286579	MN133241	21°30’17.65"N	This study	
MBM286580	MN133242	108°13’37.07"E	This study	
Loimia ingens (Grube, 1878)	China, Linqiangshidao Island	MBM286597	MN133246	21°04’46.46"N	This study	
MBM286598	MN133248	109°06’19.97"E	This study	
China, Weizhoudao Island	MBM286600	MN133244	21°30’17.65"N	This study	
MBM286601	MN133245	108°13’37.07"E	This study	
MBM286602	MN133247		This study	
Loimia arborea Moore, 1903	China, Yellow sea	MBM286582	MN133249	36°59’45.60"N 122°59’31.20"E	This study	
MBM286581	MN133250	30°59’28.59"N 122°20’39.71"E	This study	
Canada, British Columbia, Queen Charlotte Sound	BC016	HM473449	51°21’36.00"N 128°35’24.00"W	Carr et al. (2011)	
Loimia bandera Hutchings, 1990	China, Taiwan Strait	MBM286583	MN133251	25°50’26.88"N 120°14’50.28"E	This study	
MBM286584	MN133252		This study	
Loimia medusa (Savigny, 1822)	–		AY040704	–	Siddall et al. (2001)	
USA, Virginia, Virginia Beach County, Lower	SERCINVERT1643	MK308193	36°55’21.36"N 76°4’21.36"W	Direct Submission	
Loimia ramzega Lavesque et al., 2017	French, Brittany, English Channel	MNHN-IA-TYPE 1788	KY555061	48°37’37.20"N 4°34’08.50"E	Lavesque et al. (2017)	
MNHN-IA-TYPE 1789	KY555062	
MNHN-IA-TYPE 1790	KY555063	
Loimia sp.	Thailand, Phuket		AF342685	–	Colgan, Hutchings & Brown (2001)	
India, Vellar estuary	VB-2018	MG251651	–	Direct Submission	
India, Goa	GP0285	KX525508	15°34’12.00"N 73°44’24.00"E	P. Rengaiyan, B.S. Ingole, and R.R. Meena, 2016, unpublished data	
GP0286	KX525509	
GP0287	KX525510	
GP0288	KX525511	
Pista cristata Müller, 1776	Norway, Trondhejmsfjord	H88	MG270116	–	Eilertsen et al. (2017)	
Terebella lapidaria Linnaeus, 1767	United Kingdom	SIO:BIC:A1102	JX423771	50°21.00’N 4°07’48.00"W	Stiller et al. (2013)	

Molecular data and analysis

DNA was extracted from 17 individuals collected from various coastal areas of Chinese seas that represented five morphologically distinct species of the genus Loimia (Table 1). Total DNA was extracted with the DNeasy® Blood and Tissue Kit (Qiagen, Hilden, Germany) and stored at −20 °C. Partial fragments (approximately 700 bp) of the COI gene were amplified by the polymerase chain reaction (PCR) using primers LCO1490/HCO2198, LCO1490/CO1-E, polyLCO/polyHCO, and polyLCO/polyshortCOIR. The primer information is given in Table 2. Amplifications were carried out in a reaction mixture containing 2 µl of template DNA, 12.5 µl of Premix Taq™ (Takara, Otsu, Shiga, Japan), 0.5 µl of each primer (stock concentration, 10 mM), and sterile distilled H2O to a total volume of 25 µl with cycling conditions as follows: initial denaturation at 94 °C for 10 min, followed by 35 cycles of denaturation at 94 °C for 30 s, annealing at 45 °C for 40 s, and extension at 72 °C for 30 s. A final extension at 72 °C for 10 min was included. PCR-products generating distinct bands after electrophoresis on 1% agarose gels were sent to the Qingke Laboratory (Qingdao, China) for sequencing using the same set of primers that was used for PCR. Fragments with overlapping sequences (forward and reverse) were merged into consensus sequences using CONTIG EXPRESS (a component of Vector NTI Suite 6.0, Life Technologies, Carlsbad, CA, USA). The assembled sequences were checked by searching BLAST in GenBank to ensure that the DNA was not contaminated.

Table 2 Primer information.

Information of primers used for amplification and sequencing.

Gene	Primer	Sequence (5′–3′)	References	
COI	LCO1490	GGTCAACAAATCATAAAGATATTGG	Folmer, Black & Hoeh (1994)	
HCO2198	TAAACTTCAGGGTGACCAAAAA ATCA	Folmer, Black & Hoeh (1994)	
CO1-E	TATACTTCTGGGTGTCCGAAGAATCA	Carr et al. (2011)	
PolyLCO	GAYTATWTTCAACAAATCATAAAGATATTGG	Carr et al. (2011)	
PolyHCO	TAMACTTCWGGGTGACCAAARAATCA	Carr et al. (2011)	
PolyshortCOIR	CCNCCTCCNGCWGGRTCRAARAA	Carr et al. (2011)	

In addition to the sequences obtained by PCR, we downloaded all COI gene sequences of Loimia species from GenBank. COI gene sequences of two species from the family Terebellidae, namely Pista cristata Müller, 1776 and Terebella lapidaria Linnaeus, 1767 were also downloaded as the outgroups for phylogenetic analysis (Table 1).

Molecular data, including 31 sequences of the COI gene, were aligned using MUSCLE 3.8 (Edgar, 2004). Highly divergent and poorly aligned sites were omitted from the alignment according to Gblocks 0.91b (Castresana, 2000). The best-fitting nucleotide base substitution model (GTR+F+I+G4) for the alignment data was determined in IQTREE with ModelFinder (Kalyaanamoorthy et al., 2017), and a maximum likelihood tree was constructed using IQTREE (Nguyen et al., 2015) with 1,000 bootstrap reiterations (Hoang et al., 2018). A Bayesian inference tree was constructed using MrBayes 3.2 (Huelsenbeck & Ronquist, 2001). Markov chains were run for 10,000,000 generations and sampled every 100 generations. The first 25% of trees were discarded as burn-in, and the remaining trees were used to construct the 50% majority-rule consensus tree and to estimate posterior probabilities. Genetic distances were calculated using the Kimura’s two-parameter model in MEGA 7.0 (Kumar, Stecher & Tamura, 2016). Finally, all the sequences obtained in this study were submitted to GenBank (Table 1).

Zoobank registration

The electronic version of this article in portable document format will represent a published work according to the International Commission on Zoological Nomenclature (ICZN), and hence the new names contained in the electronic version are effectively published under that Code from the electronic edition alone. This published work and the nomenclatural acts it contains have been registered in ZooBank, the online registration system for the ICZN. The ZooBank Life Science Identifiers (LSIDs) can be resolved and the associated information viewed through any standard web browser by appending the LSID to the prefix http://zoobank.org/. The LSID for this publication is: urn:lsid:zoobank.org:pub:06667411-789A-49C9-AC12-8DA2858341E9. The online version of this work is archived and available from the following digital repositories: Peer J, PubMed Central, and CLOCKSS.

Results

Taxonomy

Family Terebellidae Johnston, 1846

Genus Loimia Malmgren, 1866

Type species: Loimia medusa (Savigny, 1822)

Diagnosis. Eyespots, if present, at basal part of prostomium; lobes on segments 1 and 3 or 1 and 2/3 (in combination of segment 2 and 3), sometimes also on segment 4. Three pairs of branching branchiae, on segments 2–4. Rectangular or trapezoidal mid-ventral shields from segments 2–3 to posterior region where notopodia terminate; the last segments of the glandular region usually subdivided into transverse bands. Conical to rectangular notopodia beginning on segment 4, extending for 17 segments, until segment 20; notochaetae all narrowly-winged. Neuropodia beginning from segment 5, bearing pectinate uncini, arranged in single rows on segment 5–10 and in double rows on segments 11–20. Genital papillae on segments 6–8. Pygidium smooth to papillate (Carrerette & Nogueira, 2015; Nogueira, Hutchings & Carrerette, 2015).

Remarks. Loimia species are normally found in tropical areas, with only a few species inhabiting subtropical areas, and no species inhabiting Arctic areas (Lavesque et al., 2017). The genus Loimia can be distinguished by the presence of pectinate uncini, a feature unique to Terebellidae (Nogueira, Hutchings & Carrerette, 2015). Other characteristics useful in the identification of species within this genus are: (1) the shape and position of the lobes on segments 1 and 3; (2) the number, arrangement, and shape of the mid-ventral glandular shields; and (3) the number and morphology of uncinial teeth from both anterior and posterior regions (Hutchings & Glasby, 1988, 1995; Hutchings, 1997; Londoño-Mesa & Carrera-Parra, 2005; Londoño-Mesa, 2009; Carrerette & Nogueira, 2015).

Loimia borealis sp. n.

Figs. 1–4

Figure 1 Live specimens of Loimia borealis sp. n.

(A and D) Paratype, MBM286591; (B and C) paratype, MBM286593. (A) Dorsal view; (B) ventro-lateral view; (C) detail of the anterior part, ventro-lateral view; (D) anterior part, detail of the branchiae, dorsal view. Scales: (A and B) 10 mm; (C and D) 5 mm.

Figure 2 Details of Loimia borealis sp. n.

(A–C) Holotype, MBM286585; (D–F) specimen, MBM199302. (A) Anterior part, dorsal view; (B) anterior part, lateral view; (C) anterior part, ventral view, detail of ventral shields; (D) anterior part, dorsal view; (E) anterior part, lateral view; (F) anterior part, ventral view, detail of ventral shields. Numbers refer to segments; br1, br2, br3 refer to the three pairs of branchiae; arrows point to nephridial papillae. Abbreviations: ll = lower lip, ul = upper lip, br = branchiae. Scale: 2 mm.

Figure 3 Details of Loimia borealis sp. n. under SEM. (Paratype, MBM286592).

(A) Anterior part, ventral view; (B) anterior segments, lobes on segments 1 and 3, ventral view; (C) detail of the lobe on segment 3, lateral view; (D) notopodium from segment 18; (E) notochaetae from segment 5; (F) notochaetae from segment 12; (G) uncini from posterior segment; (H) uncini from segment 12, double rows of uncini; (I) uncini from an anterior segment; (J) uncini from an anterior segment; (K) notochaetae from segment 9; (L) notopodium from segment 8. Numbers refer to segments. Abbreviations: ul = upper lip. Scales: (A) 2 mm; (B) 1 mm; (C) 500 µm; (D and L) 200 µm; (E and I) 20 µm; (F, H and K) 50 µm; (G) 15 µm; (I) 30 µm.

Figure 4 Uncini and notochaetae of Loimia borealis sp. n. (Holotype, MBM286585).

(A) Uncini, segment 20, double rows of uncini; (B) uncini, segment 6; (C) uncini, segment 95; (D) uncini, segment 21; (E) notochaetae, segment 18; (F) notopodium of the segment 8; (G) uncini, segment 5; (H) notochaetae, segment 7; (I) notopodium of the segment 7; (J) notochaetae, segment 20; (K) notochaetae, segment 8. Scales: (A and H–J) 100 μm; (B and C, E and K) 50 μm; (D and G) 20 μm; (F) 200 μm.

Materials examined. Holotype: MBM286585, mud-flat, Yangkou Village, Shandong province (37°16′34.00″N 119°02′19.44″E), coll. W. N. Wang, December 09, 2017. Paratypes: MBM286586–MBM286593, eight specimens, same collection data as holotype.

Additional material examined. MBM199208, Cangkou, Qingdao (36°11′28.44″N 120°22′55.40″E), intertidal, August 21, 1958. MBM199302, Cangkou, Qingdao (36°11′28.44″N 120°22′55.40″E), intertidal, May 22, 1959.

Description of holotype. Complete specimen with 104 segments, 82.8 mm long; thorax 13.2 mm long. Body salmon or reddish flesh-color in live specimens, with darker, yellowish mid-ventral shields terminating with blood red region (Figs. 1A–1C). Preserved body whitish, without distinct patterns of pigmentation (Figs. 2A–2C). Anterior segments compact, with posterior thoracic segments longer than anterior thoracic segments (Fig. 1B); body widest over mid posterior thorax, tapering gradually over abdominal segments (Fig. 1B). Anterior segments inflated dorsally (Fig. 1B). Abdomen with irregular swellings (Fig. 1A). Prostomium at base of dorsal surface of upper lip; basal part without eyespots; distal part shelf-like from which long and grooved buccal tentacles originate (Figs. 2B and 2C). Peristomium restricted to lips; upper lip with long free edge, projecting forward, spoon-like, shorter than wide (Figs. 2B, 2C and 3B); thicker at the base. Lower lip short, button like, partly hidden by the first pair of lateral lobes. Segment 1 dorsally narrow, with pair of well-developed lobes originating ventro-laterally; lobes with oblique dorsal margins inserted laterally to first pair of branchiae; distally rounded, roughly circular, reaching around mid-length of upper lip, mid-ventrally connected by membrane (Figs. 2B, 2C, 3A and 3B). Segment 2 reduced, conspicuous laterally and dorsally, covered by lobes of segment 3 laterally and partially fuzed to it mid-ventrally. Segment 3 with a pair of circular lobes, twice width of notopodia, originating dorso-laterally, ventral edges fuzed to upper corners of the first mid-ventral shield, dorsal margins inserted at level of dorsal edges of notopodia of segment 4, tips extending dorsally, reaching bases of branchiae (Figs. 2B and 2C). Lobes of segment 4 absent (Figs. 3A–3C). Three pairs of branched branchiae present on segments 2–4, all equal in size, inserted progressively more laterally (Fig. 2A). Branchiae with short, thick stalks and slender dendritic filaments branching dichotomously from secondary stems, originating in a spiral from short basal stems (Fig. 1D). Genital papillae small, at base of notopodia of the segments 6–8 (Figs. 2A and 2B). Seven ventral shields compact, beginning from segment 2 to segment 10; those of segments 2–4 almost fuzed into single crenulated structure, wider than those of following segments, then all about the same size (Figs. 2B and 2C); anterior shields rectangular with length five times the width, trapezoidal on segment 10; shields transversally divided on segments 10, from segment 11, shields replaced by mid-ventral groove extending posteriorly (Figs. 2B, 2C, 3A and 3B); blood red region on segments 10–13 (Figs. 1B and 1C). Notopodia extending from segment 4–20 for 17 segments, bearing two types of notochaetae within a fascicle (Figs. 3D, 3L, 4F and 4I): long chaetae narrowly bilimbate (Figs. 3E, 3F, 3K, 4E, 4H and 4K) and short smooth capillaries (Fig. 4J). Neuropodia starting from segment 5, as glandular ridges slightly raised from surface of thorax until segment on which notopodia terminate, thereafter as elongate pinnules (Figs. 2B, 2C, 2F and 3A). Uncini pectinate, with 5–6 teeth, higher than long, with oblique and concave base presenting slight protuberance; short triangular heel directed backwards for the attachment of the long, ligamental filament, with prow downwardly directed, aligned with line of teeth, connected to long filament (Figs. 4A–4D and 4G); under SEM, series of teeth with lateral fringe of minute teeth (Figs. 3G–3J). Uncini arranged in single row until segment 10, in double rows from segment 11–20, in back to back position; in single rows from segment 21 to the pygidium; abdominal uncini similar in shape but smaller in size than thoracic ones. Anus surrounded by about 7 rounded papillae. Tube with an inner lining of mucus covered by small stones and shell fragments.

Variation. The number of segments in complete specimens varies from 95 to 104. The number of ventral shields varies from 7 to 9. Genital papillae absent in some specimens. In addition, other morphological characteristics used for comparison are provided in Table 3.

Table 3 Morphological variations within the type-series of L. borealis sp. n.

Characters used to compare and morphological variations within types series of L. borealis sp. n. Abbreviations: L = length; W = width; A = absen.

	Holotype	Paratype 1	Paratype 2	Paratype 3	Paratype 4	Paratype 5	Paratype 6	Paratype 7	Paratype 8	
	MBM286585	MBM286586	MBM286587	MBM286588	MBM286589	MBM286590	MBM286591	MBM286592	MBM286593	
Oversize (L/M, mm)	82.8/13.2	77.3/11.9	73.3/13.2	72.7/6.9	59.6/12.0	80.5/7.8	63.2/9.7	63.5/10	72.7/11.5	
Number of segments	104	101	104	95	85 (broken)	90	88 (broken)	55 (broken)	94	
Eyespots	A	A	A	A	A	A	A	A	A	
Ventral shields (numbers)	8, all equal size, 4 last segments divided into 2 or more transverse lines	7, all equal size, segment 2 and 3 fuzed, 3 last segments divided into 2 or more transverse lines	7, all equal size, segment 2 and 3 fuzed, 2 last segments divided into 2 or more transverse lines	8, all equal size, segment 2 and 3 fuzed, 3 last segments divided into 2 or more transverse lines	7, all equal size, segment 2 and 3 fuzed, 2 last segments divided into 2 or more transverse lines	7, all equal size, segment 2 and 3 fuzed, 3 last segments divided into 2 or more transverse lines	7, all equal size, segment 2 and 3 fuzed, 2 last segments divided into 2 or more transverse lines	7, width gradually shorter, segment 2 and 3 fuzed, 2 last segments divided into 2 or more transverse lines	9, all equal size, segment 10 divided into 2 shields, 2 last segments divided into transverse lines	
Genital papillae (segments)	6–8	6–8	6–8	–	–	–	–	6–8	6–8	
Number of teeth (anterior × posterior uncini)	5–6 × 6	4–5 × 4–5	5–6 × 5–6	6 × 6	5–6 × 5–6	5–6 × 5–6	6 × 5–6	5–6 × 5	5–6 × 5–6	

Remarks. Loimia borealis sp. n. is characterized by its compact rectangular mid-ventral shields, each with a length five times the width, which to our knowledge are not found in any other species in the genus Loimia. Loimia arborea and L. bandera also have slender ventral shields, but they differ from L. borealis sp. n. in the shape of their broad prominent prostomia, which almost surround the mouth (Figs. 5A, 5C and 5E), and their round glandular tubercles, which run in longitudinal ridges on anterior segments near notopodia (Figs. 5B, 5D and 5F) (Hutchings, 1990; Carrerette & Nogueira, 2015; Moore, 1903). Similar to L. borealis sp. n., Loimia ingens has large round lobes on segment 3 that originate dorso-laterally and align to the base of notopodia of segment 4 (Figs. 5H and 5J). However, L. borealis sp. n. has three pairs of equal-sized branchiae, which distinguish it from L. ingens (the first pair is the longest, and the third pair is the shortest) (Fig. 5I). These species are compared in Table 4.

Figure 5 Photographs of Loimia arborea, Loimia bandera and Loimia ingens.

(A) Loimia arborea, MBM286581, specimen in life; (B–D) Loimia arborea, MBM286582; (E and F) Loimia bandera, MBM286594; (G) Loimia ingens, MBM286598; (H and I) Loimia ingens, MBM286600. (A) Posterior part lost, vento-lateral view; (B) complete, dorsal-lateral view; (C) anterior part, ventral view; (D) anterior part, detail of the glandular ridge, dorsal view; (E) posterior part lost, lateral view; (F) anterior part, detail of the glandular ridge, dorsal view; (G) live specimen, ventro-lateral view; (H) incomplete, ventral view; (I) incomplete, dorsal view; (J) anterior part, lobes on anterior segments, tentacles with brown transverse bands, ventral view. Numbers refer to segments. Abbreviations: ul = upper lip. Scales: (A–C and E–F) 1 cm; (D) 500 mm; (G–I) 2 mm; (J) 1 mm.

Table 4 List of species of Loimia used for comparison.

A list of Loimia species used for comparison with main morphological characters used for the identification. Sources: Grube (1878); Moore (1903); Caullery (1944); Hutchings & Glasby (1988); Hutchings (1990); Hutchings & Glasby (1995); Londoño-Mesa & Carrera-Parra (2005); Carrerette & Nogueira (2015). Abbreviations: L = length; W = width; A = absent; P = present.

	Type locality	Size (L/M, mm)	Eyespots	Prostomium	Lateral lobes, segment 1	Lateral lobes, segment 3	Shape of branchiae	Ventral shields (segs and shape)	Uncini (number of teeth)	
L. bandera	Pacific Ocean, Hong Kong	30 /3	A	Compact with expanded anterior lip with convoluted margins	Originating laterally as rounded elongate lobes with extend ventrally as an elevated collar connected mid-ventrally	Recurved free lateral lobes fuzed to dorsal-lateral margin of segment 3 and almost fuzed onto anterior margin of segment 4	Decreasing in size from segment 2–4, with short thick main stalk and dichotomous branching secondary stalks; terminal branches short, fine and spirally	3–12, rectangular	5–6	
L. arborea	Suruga bay, Japan	120/10	A	Broad, prominent, almost surrounding the mouth	Conspicuous rounded lobes ending on the dorsal surface with a pair of smooth rounded eminences.	Prominent dorso-lateral lobes fuzed to segment 2 and partly to segment 3	The first the largest, and the third slightly the smallest, all arborescent with about 6 main branches from a central arrangement, forming a conical shape.	3–11, rectangular becoming rapidly smaller as the tori extend	6–7	
L. ingens	SE Asia, Philippines	30 /5	A/P	Compact, collar like	Rounded lobes fuzed to each other mid-ventrally, forming a collar across dorsum	Prominent and rounded lobes fuzed laterally with segments 2 and 3	The first and second pair about the same size, the third the smallest; each with long basal stem, dendritic branching subdistally, distally loosely spiraled filaments	3–14, rectangular to trapeziform	3–7	
L. medusa	Red Sea	32/2.1	P	Compact, collar like	Large, anteriorly directed ventrally and forming scoop ventrally	Large, ear-shape, not extending ventrally; not sure derived from segment 2 and 3 or just 3	The first pair very long, about 3 times body width; the third the smallest; each with main stem and many short dendritic branches	3–12,
rectangular and trapeziform	4–5	
L. montagui	Philippine Islands	35–50/–	A	Large in size, offering a wide cross-sectional black stripe at its base	Foliaceous lobes, with rounded end, connecting each other by a membrane	Low, rounded end	Decreasing in size from segment 2–4, short branchial filaments branching dichotomously from secondary stems originating from short basal stems	3–14, first six rectangular, last two trapeziform	5–6	

Etymology. “Borealis” means northern. The species is named after its location on the northern coast of China.

Type locality. Shouguang City, Shandong Province, China (37°16′8.28″N 119°01′35.19″E).

Distribution. Northern coast of China.

Loimia macrobranchia sp. n.

Figs. 6–8

Figure 6 Photographs of Loimia macrobranchia sp. n.

(A–E) Holotype, MBM286579; (F and G) specimen, MBM199304. (A) Complete animal, alive, ventral view; (B) anterior part, dorsal view; (C) anterior part, lateral view; (D) anterior part, ventral view; (E) anterior part, lateral view, detail of branchiae; (F) anterior part, lateral view; (G) anterior part, ventral view, detail of ventral shields. Numbers refer to segments; br1, br2, br3 refer to three pairs of branchiae; unspecified arrows point to nephridial papillae. Abbreviations: ul = upper lip. Scales: (A) 1 cm; (B–G) 2 mm.

Material examined. Holotype: MBM286579, Fangchenggang City, Guangxi province, China (21°30′17.65″N 108°13′37.07″E), intertidal, coll. J. X. Sui, October 17, 2016. Paratype: MBM286580, same collection data as holotype, specimen incomplete. MBM286581, same collection data as holotype, specimen complete.

Additional material examined. MBM199304, Shatian Port, Hepu Country, Guangxi Province, China (21°31′15.55″N 109°39′1.41″E), intertidal, April 21, 1963.

Description of holotype. Specimen complete, with 125 segments, 358.9 mm long; thorax 15.7 mm long. Body yellowish brown in live material, with greenish branchiae and yellowish inclusions. Darker, reddish and yellowish mid-ventral shields terminating with blood red glandular region subdivided as transverse bands (Fig. 6A). Preserved body whitish, without distinct patterns of pigmentation (Figs. 6B–6E). Posterior thoracic segments longer than anterior segments; body swollen anteriorly, widest over mid posterior thorax, tapering gradually over abdominal segments (Fig. 6A). Notopodia beginning from segment 4 followed by 17 chaetigerous segments. Prostomium attached to dorsal surface of upper lip; basal part without eyespots; distal part almost straight, from which buccal tentacles originate. Prostomium broad, almost surrounding the mouth (Fig. 6D); upper lip with long free edge, projecting forward, shorter than wide, thrown into several vertical folds with the margins corrugated (Figs. 6C and 6D). Peristomium ventrally and laterally with a conspicuous free anterior border, ending above in a pair of narrow lobes originating dorsal-laterally and reaching around 1/3 of the upper lip length, mid-ventrally connected by membrane (Figs. 6C and 6D). Lower lip small, button like to rectangular, partly hidden by the membrane the first pair of lateral lobes (Fig. 6D). Segment 2 reduced, dorsally conspicuous, fuzed with segment 3, laterally forming a pair of small and triangular lobes (Fig. 6C). Lobes with bases wider than tips, dorsal margins aligned and fuzed to the notopodia of segment 4 (Figs. 6C and 6D). Segment 4 without lateral lobes. Three pairs of branchiae on segments 2–4 with yellowish inclusion (Figs. 6B and 6C). First pair of branchiae large, about 3 times as large as subsequent ones, with stout main stem and about 18 dendritic branches arranged in two levels on the main stem (Fig. 6E). The other two pairs of branchiae being subequal. Genital papillae small, at base of notopodia of the segments 6–8 (Fig. 6B). Ten ventral shields, starting from segment 2 to segment 12 (Fig. 6A); the first shield on segments 2–4 completely fuzed into single crenulated structure, being twice as long as following segments (Fig. 6D); anterior shields rectangular, all about the same size (Fig. 6D); behind segment 11, the glandular region subdivided as transverse bands, blood-red in live material (Fig. 6A). Bands on segment 12 and 13 trapezoid, width being twice as long as anterior shields; on segment 14 and 15 the largest, almost oval in outline; on segment 16 smaller than that on 15, oval in outline; subsequently bands becoming shorter and narrower, disappearing by segment 19 (Fig. 6A). Notopodia from segment 4, extending for 17 segments (Figs. 6A, 6C and 6D); first 8 pairs of podia long, cylindrical, following ones slender, shorter than anterior ones, almost inconspicuous. Notochaeta of two types (Figs. 7A and 7D), widely-winged, long thick chaetae in both rows throughout (Figs. 7B, 7E, 7F, 8A and 8D) and short smooth capillaries (Fig. 7C and 8B). Neuropodia starting from segment 5, as glandular ridges slightly raised from surface of thorax until segment on which notopodia terminate, thereafter as elongate pinnules. Uncini pectinate, with 5–6 teeth, higher than long, with oblique and concave base, short triangular heel directed backwards for the attachment of the ligamental filament, with prow downwardly directed, aligned with line of teeth, connected to long filament (Figs. 7G–7J, 8C, 8E–8G). Uncini arranged in single row until segment 10, in double rows from segment 11–20, in back to back position; in single rows from segment 21 until the pygidium; abdominal uncini similar in shape but smaller in size than thoracic ones. Anus rounded, without any anal papillae (Fig. 6A). Tube with an inner lining of mucus covered by sand.

Figure 7 Uncini and notochaetae of Loimia macrobranchia sp. n. (Holotype, MBM286579).

(A) Notopodium of the segment 8; (B) notochaetae, segment 5; (C) notochaetae, segment 6; (D) notopodium of the segment 18; (E) notochaetae from segment 18; (F) notochaetae from segment 11; (G) uncini, segment 21; (H) uncini, segment 5; (I) uncini, segment 15; (J) uncini, segment 53. Scales: (A) 250 μm; (B–C and E–G) 100μm; (D) 200 μm; (H–J) 50 μm.

Figure 8 Uncini and notochaetae of Loimia macrobranchia sp. n. under SEM. (Holotype, MBM286579).

(A) Notopodium from segment 6; (B) notochaetae from segment 18; (C) uncini, segment 17, double rows of uncini; (D) notopodium from segment 17; (E) uncini from segment 30; (F) uncini from segment 12; (G) uncini from segment 6. Scales: (A) 40 µm; (B and D) 200 µm; (C) 50 µm; (E) 15 µm; (F) 10 µm; (G) 20 µm.

Variations. Three specimens were collected from Guangxi Province. The number of segments in these specimens varies from 125 to 130, with the highest number of segments in the holotype. One paratype was incomplete (length, 189 mm; thorax, 13.1 mm). It has ventral grooves shaped as a trapezoid on segment 13, as a square on segment 14, and as a long rectangle on segment 15. The first pair of branchiae are approximately five times larger than the subsequent ones. The other paratype is complete (length, 125 mm; thorax, 8 mm), with relatively short broken dendritic branchiae. The stems of the first pair of branchiae are slightly thicker than the subsequent ones.

Remarks. Loimia macrobranchia sp. n. is characterized by the shape and size of its first pair of large branchiae, which are approximately three to five times larger than subsequent ones, with a thick main stem and approximately 18 dendritic branches arranged in two levels. It is easy to identify Loimia macrobranchia sp. n. by its long length of 360 mm (L. borealis sp. n. 84 mm; L. ingens 5 mm; L. bandera 30 mm; L. arborea 120 mm). The largest species within this genus is Loimia ramzega Lavesque et al., 2017 from French waters (maximum length, 650 mm), and it differs from L. macrobranchia sp. n. in the shape of its first pair of branchiae, with a thick main stem and many dendritic branches arranged in five levels (Lavesque et al., 2017).

Etymology. We attribute to this taxon the epithet “macrobranchia” to indicate the first pair of large branchiae.

Type locality: Guaishitan Beach, Fangchenggang City, Guangxi province, China (21°30′4.44″N 108°12′59.81″E).

Distribution: China, Guangxi Province.

Phylogenetic analysis

Kimura’s two-parameter pairwise genetic distance between L. arborea from this study and that from GenBank (Carr et al., 2011) was 32.1%, suggesting that they are different species. The genetic distance between Loimia sp. from India and that from Thailand was 0.6%, indicating that they are identical species. Except for Loimia sp. and L. arborea, the mean sequence divergences within species ranged from 0.0% (L. bandera and L. ramzega) to 0.4% (L. macrobranchia sp. n.), and the mean sequence divergences between species, excluding the outgroups, ranged from 15.4% (L. ingens and Loimia sp.) to 30.2% (L. arborea from GenBank and L. ingens). The highest divergence between Loimia species and the outgroups was 29.8% (L. borealis sp. n. and T. lapidaria).

The combined phylogenetic trees (Fig. 9) reconstructed from both maximum likelihoods (ML) and Bayesian inference (BI) analyses were generally comparable, and all ingroup taxa were recovered in a clade with high support value (BP (maximum likelihood bootstrap percentage) > 99%, PP (Bayesian posterior probability) = 1.0). Loimia arborea, whose sequence was downloaded from GenBank (Carr et al., 2011), clustered together with the outgroups (T. lapidaria and P. cristata) with moderately high support (BP = 73, PP = 0.99). In addition, the average genetic divergence between L. arborea species from GenBank and other L. arborea species from this study was 32.1%, indicating that they are different species.

Figure 9 Phylogenetic tree obtained by the Maximum likelihood analysis based on the COI gene sequences.

Maximum likelihood bootstrap scores (BP, left) and Bayesian posterior probabilities (PP, right) are indicated adjacent to each node. Values of BP ≥ 75% and PP ≥ 0.95 are marked in bold. Only values of BP ≥ 50% and PP ≥ 0.5 are shown. Species names and sequence origins are given next to the corresponding clades. Additional details on the sequences used for this tree are given in Table 2. Clade annotations A and B are discussed in the text.

Phylogenetic tree reconstruction of the COI gene showed two well supported main clades (Fig. 9, clades A, B) with moderately high support (BP = 73, PP = 0.99). Clade A included L. ingens from China, Loimia sp. from Thailand and India, L. medusa from the USA, and L. ramzega from France. Loimia ingens and Loimia sp. formed a highly supported clade (BP = 89, PP = 1.0), indicating their close relationship. Clade B grouped L. macrobranchia sp. n., L. arborea, L. bandera, and L. borealis sp. n. (Fig. 9). Loimia arborea and L. bandera formed a strongly supported subclade (BP = 97, PP = 1.00), illustrating that they are more genetically related than the rest of the species within clade B.

Discussion

We re-identified the specimens deposited in the MBMCAS under the name L. medusa and collected new specimens near their recorded sites. This analysis revealed that the specimens belonged to two different species, both new and significantly different from L. medusa as described by Hutchings & Glasby (1988). One species, L. borealis sp. n., was collected from the Shandong Peninsula. It differed from L. medusa by the presence of large round lobes on segment 3 and compact long rectangular mid-ventral shields (Table 4). The other species, L. macrobranchia sp. n., was identified in Guangxi Province. Similar to L. medusa, the first pair of branchiae were approximately three times longer than the body width. However, L. medusa had eyespots arranged in small groups on each side of the posterior prostomium, which were absent in L. macrobranchia sp. n. (Table 4). Hutchings & Glasby (1995) designated a neotype for Loimia medusa and indicated that their distribution was restricted to the Arabian Sea. Due to the long distance between China and the type locality (Red Sea, neotype from the Persian Gulf) of L. medusa, we believe that the older records of L. medusa in China are misidentifications and this species does not occur in Chinese seas.

According to Carrerette & Nogueira (2015), Loimia montagui was initially identified as Terebella montagui from the Philippines by Grube (1878), and then assigned to the genus Loimia by Hartman (1959). Subsequently, Terebella montagui Grube, 1878 was redefined as Loimia grubei by Holthe (1986). Presently, the taxonomic significance of Loimia montagui (Grube, 1878) in WoRMS (http://www.marinespecies.org) is uncertain. Given the brief descriptions of L. grubei by Holthe (1986) and L. montagui by Hartman (1959), we referred to the descriptions of L. grubei by Carrerette & Nogueira (2015) and L. montagui by Caullery (1944). According to these descriptions, the species has a pair of round lateral lobes on segment 1, slightly shorter round lobes on segment 3, branchiae with short branchial filaments branching dichotomously from the secondary stems that originate from the short main stem, and ventral shields extending from segments 2 to 14 (Table 4). By contrast, L. borealis sp. n. has large ventral lateral lobes on segment 1, well-developed lobes on segment 3 that are slightly larger than those on segment 1, and seven mid-ventral shields from segments 2–10 (Figs. 2B and 2C). As for L. macrobranchia sp. n., the lobes on segment 1 are long, and they extend as elevated collars that connect mid-ventrally (Figs. 6C and 6D). In addition, 10 ventral shields from segments 2 to 12 are present. Previously, Hutchings & Glasby (1988) examined materials collected from Australia that were initially identified as L. montagui. Subsequently, they classified these specimens as L. ingens. As for the materials collected from coastal areas of Chinese seas and identified as L. ingens, they differed from L. montagui in the shape of their branchiae, namely in their long basal stems, subdistal dendritic branches, and loose spiral filaments (Fig. 5I).

Carrerette & Nogueira (2015) suggested that the shape and position of the lobes on segments 1 and 3, the shape of the ventral glandular shields, and the number of segments on which they are present, together with the number and general morphology of the uncinial teeth, are critical in the identification of Loimia species. As observed by us, the shape and distribution of mid-ventral shields, which have rarely been included in species descriptions (Carrerette & Nogueira, 2015), are useful characteristics in the taxonomic classification of Loimia species. The mid-ventral shield shape of Loimia species varies from rectangular to trapezoidal, which is typical of Terebellidae (Nogueira, Hutchings & Fukuda, 2010; Nogueira, Fitzhugh & Hutchings, 2013). These shields usually extend from anterior segments to posterior segments where notopodia terminate, appearing blood red-to-yellowish in live materials (Figs. 1C, 5A, 5G and 6A) and pale yellow-to-white in preserved specimens (Fig. 5B). Londoño-Mesa (2006, 2009) and Londoño-Mesa & Carrera-Parra (2005) commented on the usefulness of the distribution of ventral shields as a specific characteristic for Loimia species, although it is difficult to determine the exact anterior segment (mostly short and fuzed to each other) on which the shields begin, and the particular segment on which they end. Carrerette & Nogueira (2015) also mentioned that differences in the shape of shields was an additional useful characteristic in the identification Loimia species.

After re-identifying the specimens that have been deposited in the MBMCAS for many decades, we found that the important morphological characteristics of branchiae and lobes were destroyed, and their shape was undefinable (Figs. 2D–2F, 6F and 6G). By contrast, the characteristics of mid-ventral shields were well preserved, and they were easily recognizable. Equally-sized mid-ventral shields in L. borealis sp. n. spanned from segments 2 to 10, each with a length significantly longer than the width (Figs. 2D–2F). In old materials of L. macrobranchia sp. n. and L. ingens, the rectangular ventral shields spanned from segments 2 to 12 (Figs. 6F and 6G), similar to the shields in new materials. In agreement with Carrerette & Nogueira (2015), we conclude that the shape of mid-ventral shields is a significant character for the identification of species of Loimia.

In the phylogenetic tree (Fig. 9), Loimia macrobranchia sp. n. was recovered as a monophyletic clade, although nodal support was low. Loimia borealis sp. n. was retrieved as a sister to the clades of L. arborea and L. bandera with moderate support (Fig. 9). In addition, Kimura’s two-parameter pairwise genetic distance between L. macrobranchia sp. n. and other Loimia species ranged from 18% to 24%, and the genetic distance between L. borealis sp. n. and other species ranged from 20% to 24%, indicating genetic divergences between new species and other Loimia species.

Conclusions

This study is the first to analyze molecular sequence data of Loimia species from Chinese seas. The new species L. borealis sp. n. and L. macrobranchia sp. n. were established based on their morphological and molecular findings. Loimia borealis sp. n. is distinguished from other valid Loimia species by its compact, slender, and rectangular ventral shields, which indicates that the ventral shield shape is a significant characteristic in the identification of Loimia species. The other species, L. macrobranchia sp. n., is characterized by its large size and large first pair of branchiae. A revision of the genus Loimia from Chinese seas was made, and we suspect that older records of L. medusa and L. montagui in China are misidentifications. A key for the identification of Loimia species in Chinese seas is provided. This key only contains valid species listed in WoRMS (http://www.marinespecies.org).

Key to the species of Loimia found in China seas

1. First pair of branchiae three to five times longer than the second and third pairsL. macrobranchia sp. n.

-First pair of branchiae equal or slightly longer than the second and third pairs2

2. Lobes of segment 3 small and rectangular, dorsum with rounded glandular tubercles run together into a low longitudinal ridge on the first 10 anterior segments each side just beside the notopodia3

-Lobes of segment 3 large and semicircular, dorsum without glandular and longitudinal ridge4

3. Lateral lobes on segments 2 partly fuzed with segment 3L. arborea

-Lateral lobes on segments 2, 3 and 4L. bandera

4. Eyespot absent; mid-ventral shields all equal-sized, anterior shields rectangular, with length five times the widthL. borealis sp. n.

- Red or black eyespots, if present, on basal part of prostomium; nine mid-ventral shields gradually taper, anterior shields rectangular with length two times the width, then trapezoidal or squareL. ingens

Supplemental Information

Supplemental Information 1 COI sequences amplified from our study.

Click here for additional data file.

Supplemental Information 2 Kimura’s 2-parameter genetic distances among Loimia species collected from China and Genbank.

Click here for additional data file.

We are grateful to Dr. Xinming Liu (Institute of Oceanology, Chinese Academy of Sciences, Qingdao) and Dr. Dong Dong (Institute of Oceanology, Chinese Academy of Sciences, Qingdao) for their kind help with photographing the specimens.

Additional Information and Declarations

Competing Interests

Author Contributions

DNA Deposition

Data Availability

New Species Registration

The authors declare that they have no competing interests.

Weina Wang conceived and designed the experiments, performed the experiments, analyzed the data, prepared figures and/or tables, authored or reviewed drafts of the paper, and approved the final draft.

Jixing Sui conceived and designed the experiments, performed the experiments, analyzed the data, authored or reviewed drafts of the paper, and approved the final draft.

Qi Kou conceived and designed the experiments, authored or reviewed drafts of the paper, and approved the final draft.

Xin-Zheng Li conceived and designed the experiments, authored or reviewed drafts of the paper, and approved the final draft.

The following information was supplied regarding the deposition of DNA sequences:

The COI sequences described here are available at GenBank: MN133237 to MN133242, MN133244 to MN133252 and MT246207 to MT246208.

The following information was supplied regarding data availability:

The raw measurements are available in the Supplemental Files.

The specimens described in the article are stored in the Marine Biological Museum of the Chinese Academy of Sciences. The location of all specimens and the accession numbers for each specimen are available in Table 1.

The following information was supplied regarding the registration of a newly described species:

Publication LSID: urn:lsid:zoobank.org:pub:06667411-789A-49C9-AC12-8DA2858341E9.

Loimia borealis Wang, Sui, Kou & Li LSID: urn:lsid:zoobank.org:act:E292C1DE-FD03-4AAF-BCA0-B4BF339984A2.

Loimia macrobranchia Wang, Sui, Kou & Li LSID: urn:lsid:zoobank.org:act:CDB7AA6A-CCBD-4873-A6F6-AE867DCD8616.

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
