# Peer review of "Review of the genus Loimia Malmgren, 1866 (Annelida, Terebellidae) from China seas with recognition of two new species based on integrative taxonomy"

_PeerJ, doi:10.7717/peerj.9491_

## Round 0.1 · original submission · Major Revisions

I have heard back from two expert reviewers, both of whom have numerous comments on your work. Please consider all of their comments carefully in your responses. I would like to ask that you also include responses to all sticky notes in reviewer 2's sticky notes within your rebuttal to make it easier to assess all of your responses.

Reviewer 1 ·

Basic reporting

The language is overall clear and unambiguous but some spelling and structural issues that are highlighted in the General Comments section. I would further advise to replace "huge" and "enormous" with more toned synonyms.

The introduction should mention how Loimia is identified compared to other Terebellidae (uncini, branchiae, lobes), so that it is clear how material was assigned to this group. There are further some nomenclature inconsistencies because some of the species in Chinese seas were not originally placed in Loimia and the author name and publication year should therefore be enclosed in parentheses to follow nomenclatural rules. This applies to 3 species (L.76-77), Loimia ingens (Grube, 1878), Loimia medusa (Savigny, 1818), Loimia montagui (Grube, 1878).

Experimental design

A main issue is that not all available sequence data is being used. While sequences for two other terebellids are included as outgroups from published data, available sequences for the species under investigation are not considered. Genbank has COI, 16S and one 18S for species of Loimia (Loimia ramzega, Loimia medusa, Loimia arborea, Loimia ingens) that were not used here. If there is evidence that these sequences should not be used (e.g. suspected misidentifications of the species) that would be valid but it should be justified why publicly available data is not being used here to gain the most insight into Loimia species.

The phylogenetic analyses are carefully performed but I would encourage to add a maximum likelihood analysis of the data. The analysis should involve selecting the best fit model for each data partition (COI, 16S, 18S) rather than estimating a substitution model across all three partitions because these genes are from different genomic compartments (mitochondrial, nuclear genome) and have different rates and patterns of substitution. The partitioned dataset can then be used in a maximum likelihood framework with bootstrap supports. RAxML or IQTREE can be used for these purposes and both approaches have free-to-use servers that can be used with small datasets like this one. IQTREE even performs model selection with Modelfinder as part of the analysis.

I discourage the use of "basal" when describing phylogenetic relationships among species (albeit this term being very commonly used in the literature). Krell & Cranston 2004 Systematic Entomology describe the problem clearly: "Every branching in a (phylogenetic) tree is rotatable (see Fig. 1). Of course, the tree has a base, and there is a most basal branching and a next most basal branching, but there is no such thing as the most basal clade. Because branchings are rotatable, there are always two most basal clades (if the most basal branching is completely resolved, Fig. 1) or even more most basal clades (if the most basal branching is not completely resolved, e.g. in Polyneoptera in Fig. 1). Both branches originating from a node (i.e. the two sister groups) are of equal age and have undergone equivalent evolutionary change. Whether a group has branched off early (basal) or later in the phylogeny contains no information about this particular group, but information about both this group and its sister group, because both branched off at the same time." I encourage the authors to describe the phylogenetic relationships among Loimia species using descriptions about sister groups. This applies to the Abstract and the paragraph L.388-394, L.456-462.

Validity of the findings

no comment

Additional comments

Thank you for a carefully conducted and easy to follow study.

Minor comments on language and structure:
Abstract
- Add Annelida, Terebellidae after first mention of Loimia
- "collected from the coasts of Shandong..."
- Remove point after "respectively: Loimia."
- 2 spaces missing: "COI, 16S rRNA and 18S rRNA"
- small case on "phylogenetic tree"
- upper case on "identifying Loimia species"
- "China seas" could be better as Chinese seas

Abstract, L.264, L.266, L.363, L.460, L.468: I believe it is uncommon to start sentences with an abbreviation (of Loimia to L.)

L.79: add space after Malmgren, 1865
L.75: capitalize China
L.102: maybe better "from coastal areas of
L.114: "using the primer pairs:
L.142-143: repeated reference to Gblocks compared to L.136
L.180: maybe "convoluted" or "complicated taxonomic history" rather than "confused"
L.181: remove dot before reference
L.181: "occur in tropcial areas" or "are distributed in tropical areas"
L.183: Add "The most useful character"
L.196: Add space after August 21,
L.216: remove ")"
L.217: Add space after October 22,
L.219: lower case specimens
L.252: replace "and" with "," in Figure reference
L.277: Add "." at end of sentence.
L.281: remove ")"
L.286: remove ")", add space after April 21,
L.291: add space after August 21,
L.304: remove ")"
L.376-394: all occcurrences of L. arborea and L. bandera misspelled
L.396: maybe better "reidentified" than rechecked
L.397: remove "out" after finding, remove "very"
L.398: "new to science"
L.400: add space segment 3
L.409: misformatted reference
L.411: "does not dwell in Chinese seas"
L.411-413: sentence is difficult to understand and the last references seem misformatted.
L.414 : remove . after Loimia
L.416: we referred to the description"
L.438: "where notopodia end"
L.446: "species that have been deposited"
L.460: lower case phylogenetic
L.467: suggest to remove "independent" because COI and 16S are not independent because they get inherited together on the mitochndrial genome. Could be stronger to say " from three mitochondrial and nuclear molecular markers"
L.473: why "preliminary", remove?
L.477: add "pairs" after third to be consistent with L.479-480
L.496: remove "extremely"

Table 4: add space L. arborea and before Japan
Table 5: misspelled arborea and bandera, inconsistent spacing sp.nov.

Reviewer 2 ·

Basic reporting

The manuscript is poorly written- Intro very confusing
latest references are not quoted

the format for new species is not up to current standards for new species

Experimental design

not applicable

Validity of the findings

ok but fail to understand that molecular data from material outside of type locality is not useful

Additional comments

a major revision is needed have added sticky notes throughout the manuscript

Annotated reviews are not available for download in order to protect the identity of reviewers who chose to remain anonymous.

---

## Round 0.2 · Minor Revisions

I have heard back from one reviewer, who has found the manuscript to be well revised. There are still some small areas that need correction, and many of these pertain to English and use of words related to ICZN. Please go over all corrections and your manuscript very carefully before any resubmission. I look forward to seeing a revised manuscript.

Reviewer 2 ·

Basic reporting

see attached file still some typos and some changes still needed

Experimental design

ok

Validity of the findings

ok

Additional comments

see my attached file

Annotated reviews are not available for download in order to protect the identity of reviewers who chose to remain anonymous.

---

## Round 0.3 · Minor Revisions

Thank you for your revisions. From a scientific point of view, your work is ready to be accepted. However, in a final read over your work, I have noticed many small grammatical and similar errors, more than can be handled at the proof stage. I have edited the Introduction with tracked changes to show you the level of edits still needed to bring the English up to international standards. Please either consult with a colleague or professional service to do so, and provide me with information on the name of who checked your manuscript.

---

## Round 0.4 · accepted · Accept

Thank you for the English editing. There are still some small errors that can be corrected at the proof stage. As well, I would like the authors to ensure ALL authorities are correct by checking against the authorities listed in WoRMS - a quick perusal noted a few differences in your final submitted paper.